# Effect of Carbon Sources on Pyrite-Arsenopyrite Concentrate Bio-oxidation and Growth of Microbial Population in Stirred Tank Reactors

**DOI:** 10.3390/microorganisms9112350

**Published:** 2021-11-13

**Authors:** Aleksandr Bulaev, Aleksandra Nechaeva, Yuliya Elkina, Vitaliy Melamud

**Affiliations:** Research Center of Biotechnology of the Russian Academy of Sciences, Leninsky Ave. 33 bld. 2, 119071 Moscow, Russia; nechaevasasha709@gmail.com (A.N.); yollkina@mail.ru (Y.E.); vmelamud.inmi@yandex.ru (V.M.)

**Keywords:** biohydrometallurgy, acidophilic microorganisms, pyrite, arsenopyrite, sulfide concentrates, carbon dioxide, molasses

## Abstract

Tank bio-oxidation is a biohydrometallurgical technology widely used for metal recovery from sulfide concentrates. Since carbon availability is one of the key factors affecting microbial communities, it may also determine the rate of sulfide concentrate bio-oxidation. The goal of the present work was to evaluate the effect of carbon sources on the bio-oxidation of the concentrate containing 56% pyrite and 14% arsenopyrite at different temperatures (40 and 50 °C) in stirred tank reactors. CO_2_ was supplied into the pulp of the first reactor (about 0.01 L/min) and 0.02% (*w/v*) molasses was added to the pulp of the second one, and no additional carbon sources were used in the control tests. At 40 °C, 77% of pyrite and 98% of arsenopyrite were oxidized in the first reactor, in the second one, 73% of pyrite and 98% of arsenopyrite were oxidized, while in the control reactor, 27% pyrite and 93% arsenopyrite were oxidized. At 50 °C, in the first reactor, 94% of pyrite and 99% of arsenopyrite were oxidized, in the second one, 21% of pyrite and 94% of arsenopyrite were oxidized, while in the control reactor, 10% pyrite and 92% arsenopyrite were oxidized. The analysis of the microbial populations in the reactors revealed differences in the total number of microorganisms and their species composition. Thus, it was shown that the use of various carbon sources made it possible to increase the intensity of the concentrate bio-oxidation, since it affected microbial populations performing the process.

## 1. Introduction

Bio-oxidation of sulfide concentrates in stirred tank reactors is a biohydrometallurgical technology that allows enhancing gold recovery from refractory sulfide concentrates [1,2,3]. Bio-oxidation has advantages over other technologies (roasting, POX) and does not require high energy consumption and makes it possible to avoid toxic gas emissions, when processing concentrates containing arsenic [3,4,5]. Currently, reactor bio-oxidation accounts for about 5% of global gold production [2]. BIOX^®^, Bacox^TM^, and BIONORD^TM^ processes are examples of commercialized bio-oxidation technologies [6,7,8].

Bio-oxidation processes are carried out in aerated stirred tank reactors connected in series equipped with cooling systems [3,6,7,8]. The biohydrometallurgical processing of sulfide concentrates is based on sulfide mineral bio-oxidation by aerobic acidophilic iron- and sulfur-oxidizing microorganisms. Industrial biohydrometallurgical processes are always carried out by mixed microbial populations, which include several species of iron- and sulfur-oxidizing microorganisms [1,3]. The composition of microbial communities in bio-oxidation reactors depends on different factors, including the composition of the oxidized concentrate, temperature, pH, oxygen, and carbon availability [3].

Bio-oxidation of sulfide minerals, components of gold-bearing concentrates (arsenopyrite, pyrite), may be described by the following simplified overall reactions [6]:2FeAsS + 7O_2_ +H_2_SO_4_ + 2H_2_O→2H_3_AsO_4_ + Fe_2_(SO_4_)_3_(1)
4FeS_2_ + 15O_2_ + 2H_2_O→2Fe_2_(SO_4_)_3_ + 2H_2_SO_4_(2)

Reactor bio-oxidation may be performed at different temperatures using different microorganisms, but existing industrial processes are usually carried out at temperatures of 40–45 °C since this temperature range allows achieving maximum sulfide oxidation rates and minimize cooling requirements caused by the release of heat during the bio-oxidation of minerals [3,6]. Various thermotolerant and moderately thermophilic acidophilic microorganisms usually predominate in microbial communities formed in long-term processes of sulfide concentrate bio-oxidation: bacteria of the genera *Leptospirillum* and *Sulfobacillus*, moderately thermophilic representatives of the genus *Acidithiobacillus* (*A. caldus*), as well as archaea of the family *Ferroplasmaceae* (genera *Acidiplasma* и *Ferroplasma*) [9,10,11,12,13,14,15,16,17,18,19,20,21,22,23,24]. Despite this, long-term temperature rise in the reactors can inhibit bio-oxidation activity [3,6].

Microorganisms involved in the oxidation of sulfide minerals include both autotrophs, which fix dissolved carbon dioxide using the energy obtained by the oxidation of ferrous iron and sulfur compounds, and mixo- and heterotrophs, which require organic carbon sources for stable growth, despite they also gain energy by the oxidation of inorganic compounds [1,25,26,27]. 

Differences in the properties determine interactions between microorganisms in bio-oxidation reactors. In particular, it was shown that between autotrophs, on the one hand, and heterotrophs and mixotrophs, on the other hand, there are trophic interactions that occur both in bioleach reactors and in natural ecosystems [11,28,29]. During the growth of autotrophic acidophiles, oxidizing ferrous iron, and sulfur, exometabolites are accumulated in the medium [30,31,32,33]. Exometabolites produced by autotrophs may be consumed by mixo- and heterotrophs as a carbon source. It should be noted that some microorganisms, which predominate in the communities of bioleach reactors, are mixo- and heterotrophs (bacteria of the genus *Sulfobacillus* and archaea of the family *Ferroplasmaceae*) and their activity depends on the presence of autotrophs (bacteria of the genus *Leptospirillum* and *A. caldus*) [1,25,26,27].

Since carbon availability is important for microbial populations of bioleach reactors, it may affect bio-oxidation of sulfide concentrates [3]. In a number of works, the effect of various carbon sources on the composition of microbial communities in bioleach reactors, as well as on the rate of the bio-oxidation process was studied. For example, in [11], it was shown that the composition of microbial populations of bioleach reactors connected in series changed from the first reactor to the third one due to the increase in the concentration of dissolved organic carbon, which accumulated in the reactors due to the activity of autotrophic microorganisms.

Industrial practice demonstrated that maintaining the concentration of dissolved CO_2_ in the medium is important to maintain the required rate of the bio-oxidation process. According to the recommendations of BIOX^®^ developers, the concentrate should contain at least 2% carbonate to provide sufficient CO_2_ to promote the growth of microorganisms. If no carbonate is present in the concentrate, limestone or CO_2_ must be added to the primary reactors as a carbon source [6].

Besides carbon dioxide, the effect of organic substances, which can be a carbon source for mixo- and heterotrophic microorganisms, on the efficiency of the bio-oxidation process of sulfide concentrates was also investigated. In our previous work, it was shown that the rate of bio-oxidation of pyrite-arsenopyrite concentrate at 45 °C was influenced by the addition of 0.02% (*w*/*v*) yeast extract (YE) into the medium [20]. The introduction of an organic carbon source in this case made it possible to increase the rate of oxidation of sulfide minerals. 

Thus, the results of the studies performed show that the introduction of additional carbon sources allows intensifying bio-oxidation of sulfide concentrates and affects the composition of microbial communities. At the same time, there is not enough data in the literature that allow to analyze the effect of various carbon sources on the bio-oxidation of concentrates, as well as on the composition of microbial communities. Therefore, studies on the effect of different carbon sources on the rate of sulfide concentrate bio-oxidation and on the composition of microbial communities that formed under different conditions in order to identify groups of microorganisms that can make the greatest contribution to the bio-oxidation process are of interest.

The goal of this work was to determine the effect of an additional carbon source (CO_2_ and molasses) on the bio-oxidation of gold-containing pyrite-arsenopyrite flotation concentrate and growth of microbial population in stirred tank reactors at different temperatures.

## 2. Materials and Methods

### 2.1. Concentrate

The composition of the concentrate is shown in Table 1. The main sulfide minerals of the concentrate were pyrite (56%) and arsenopyrite (14%). In our previous work, it was shown that the concentrate is refractory since gold recovery by direct cyanidation did not exceed 57.4%. It was shown that bio-oxidation in a continuous mode at 40 °C and residence time of 6 days made it possible to oxidize 70% of sulfide sulfur and increase the rate of gold recovery by cyanidation up to 83% [34].

### 2.2. Experimental Setup and Biooxidaton

In the present work, concentrate bio-oxidation was carried out in a batch mode in 2.5 L reactors under the following conditions: aeration—5 L/min, 500 rpm, the temperature in the first experiment was 39–40 °C, and in the second one, it was 49–50 °C, the pulp density (solid to liquid ratio, S:L) was 1:5 (200 g of the concentrate per 1000 mL of liquid medium), the duration of the experiment was 40 days. The temperature in the reactors was maintained using ELMI TW-2.03 circulating water baths (Elmi, Riga, Latvia) and U-shaped titanium heat exchangers.

For the experiments, we used a liquid nutrient medium containing mineral salts (g/L): (NH_4_)_2_SO_4_—0.75, KCl—0.05, MgSO_4_ × 7H_2_O—0.125, K_2_HPO_4_—0.125, distilled water—1.0 L, which was successfully used in our previous works for sulfide concentrate bio-oxidation [26,34]. The initial pH was adjusted by adding 5 mL/L of 98% chemically pure sulfuric acid to the medium. After adding the concentrate to the medium, the pulp was incubated for 1 day without inoculum to stabilize the pH.

To determine the effect of additional carbon sources on the bio-oxidation, carbon dioxide and molasses were used. CO_2_ was fed into the pulp of the first reactor (approximately 0.01 L/min) and 0.02% molasses (*w/v*) was added to the pulp of the second reactor (at the beginning of the experiment, on days 10, 20, and 30). No additional carbon sources were added to the control reactor.

A microbial population formed during continuous bio-oxidation of the same sulfide concentrate at 40 °C was used as inoculum, in which acidophilic bacteria *Leptospitillum ferriphilum*, *Sulfobacillus* spp., as well as archaea *Ferroplasma acidiphilum* and *Acidiplasma* sp. [24,34] were predominant. The inoculum was introduced into the reactors in such a volume that the initial total number of microbial cells in the liquid phase was ~1 × 10^8^ cells/mL.

### 2.3. Sampling and Analysis

To analyze the activity of the bioleaching, samples of the liquid phase were collected every 5 days. In all samples, pH, and redox potential (Eh) were determined using a pH-150MI pH meter (Izmeritelnaya tekhnika, Moscow, Russia), ferrous and ferric iron and arsenic concentrations were measured by trilonometric and iodometric titration, respectively [35,36]. Quantitative assessment of microorganisms was carried out by direct counts using an Amplival (Carl Zeiss, Jena, Germany) microscope equipped with a phase-contrast device.

After the bio-oxidation, the solid residues of bio-oxidation were separated from the liquid phase of the pulp, dried, and analyzed to determine the oxidation state of sulfide minerals. Determination of the content of iron, arsenic, and sulfur was carried out using phase analysis methods [37]. The mineral compositions of the concentrate and bio-oxidation residues were determined by X-ray diffraction using a DRON-2 diffractometer (Burevestnik, St. Petersburg, Russia).

### 2.4. Microbial Population Analysis

The analysis of the composition of microbial communities that formed under the experimental conditions was carried out by high-throughput sequencing on the MiSeq system (Illumina, San Diego, CA, USA). For analysis, pulp samples were taken from bio-oxidation reactors (25 mL) at 20, 30, and 40 days of the experiment. The biomass from the liquid phase of the pulp was collected using an Allegra X-22 centrifuge (Beckman Coulter, Indianapolis, IN, USA). To collect the biomass from the pulp sample, the solid phase was first separated by centrifugation at 1000 rpm (103× *g*), and then the biomass was precipitated from the supernatant by centrifugation at 9500 rpm (9299× *g*). Biomass preparation, DNA isolation, library preparation based on the V3–V4 region of the 16S rRNA gene, amplicon preparation, sequencing on the MiSeq system (Illumina, San Diego, CA, USA) were performed as described previously [23,38].

### 2.5. Data Processing

Chemical analyses used in the work were performed in duplicate. Processing of the results was carried out using the MS 15.0.459.1506 Excel 2013 software (Microsoft, Redmond, WA, USA).

## 3. Results

### 3.1. Biooxidation under Mesophilic Conditons

The results of the experiment on the concentrate bio-oxidation at 40 °C are shown in Figure 1 and Figure 2, as well as in Table 2 and Table 3.

#### 3.1.1. Liquid Phase Analysis

Figure 1 shows changes in the liquid phase parameters at 40 °C. The presented curves demonstrate that, at 40 °C, carbon dioxide influenced the bio-oxidation to a greater extent than the addition of molasses.

The pH values of the liquid phase in reactors 1 (carbon dioxide) and 2 (molasses) sharply decreased. After 10 days of bio-oxidation, the pH in reactor 1 decreased from 1.31 to 0.65 on the 19th day. Then, the pH in the reactor was maintained by adding CaCO_3_ to avoid inhibiting bio-oxidation due to too high acidity. After 25 days of bio-oxidation, the pH value was stabilized in the range 1.13–1.43 (Figure 1a, curve 1). In reactor 2, in which the medium was supplemented with molasses, the pH reached a minimum on the 22nd day and was 1.08, after which it was stabilized in the range of 1.11–1.37 (Figure 1a, curve 2). In reactor 3, where no additional carbon sources were used, the pH was stabilized at a level of 1.26–1.55. All reactors required CaCO_3_ addition to maintain the pH level. In reactor 1, the CaCO_3_ consumption was 314 kg/t of the concentrate, while in reactors 2 and 3 the consumption comprised 292 and 71 kg/t, respectively. 

Changes in the Eh (Figure 1b), which reflect the activity of oxidation of iron ions in the medium, corresponded to changes in pH level (Figure 1a). In reactor 1, the Eh value began to grow rapidly after 10 days and reached 755 mV on the 13th day of bio-oxidation, and then it was in the range of 730–770 mV (Figure 1b, curve 1). In reactor 2, Eh reached the value of 750 mV on the 21st day of bio-oxidation and then was in the range of 727–766 mV (Figure 1b, curve 2). In reactor 3, the Eh values during almost the entire experiment were significantly lower than in reactors 1 and 2 and did not exceed 610 mV for up to 20 days, after which Eh began to grow rapidly and by the end of the experiment, it was practically equal to the values in reactors 1 and 2 (Figure 1a).

The diagrams in Figure 1c,d show changes in the concentration of iron ions. Up to the 10th day of bio-oxidation, the total concentration of iron ions practically did not increase in all reactors (Figure 1c, curves 1–3), while the concentration of Fe^2+^ ions increased in all reactors (Figure 1d, curves 1–3). This indicates that in the first 10 days of the experiment, the bio-oxidation activity was low, and there was an accumulation of ferrous iron ions in the medium due to the chemical oxidation of the concentrate minerals by Fe^3+^ ions, which were introduced into the medium with the inoculum.
FeS_2_ + 14Fe^3+^ + 8H_2_O→15Fe^2+^ + 16H^+^ + 2SO_4_^2−^(3)
FeAsS + 5Fe^3+^ → S^0^ + As^3+^ + 6Fe^2+^(4)

After 10 days, in reactors 1 and 2, the total concentration of ferric and ferrous ions in the medium increased at a high rate (Figure 1c, curves 1 and 2). In reactor 1, the concentration was close to the maximum on the 31st day of bio-oxidation, reaching 40.60 g/L, after which it did not change significantly (Figure 1c, curve 1). In reactor 2, the rate of increase in the concentration of iron ions was lower, and the concentration was practically equal to that in reactor 1 by the end of the experiment, on day 37 (Figure 1c, curve 2). It should be noted that in reactors 1 and 2, ferrous iron in the medium was oxidized, and its concentrations became insignificant after 13 and 20 days, respectively, slightly increasing only at the end of the experiment (Figure 1d, curves 1 and 2). In control reactor 3, the total concentration of iron Fe^3+^ and Fe^2+^ ions in the medium grew slowly to 36 days from 3.60 to 7.15 g/L. After the 37th day and up to the 40th day, the concentration sharply increased from 7.15 to 19.00 g/L (Figure 1c, curve 3). At the same time, the concentration of Fe^2+^ ions in the medium in reactor 3 was higher than 1 g/L until 34 days and then decreased to trace values (Figure 1d, curve 3). 

The dynamics of changes in the concentration of arsenic in the medium differed from the dynamics of changes in the concentration of iron ions (Figure 1e). In reactors 1 and 2, the arsenic concentration began to increase after 10 days of bio-oxidation, reaching a maximum at 15th and 20th days (13.6 and 11.9 g/L), respectively (Figure 1e, curves 1 and 2). Then the concentration of arsenic in the medium fluctuated. This can be explained by the fact that during sulfide concentrate bio-oxidation, the concentration of arsenic in the medium can both increase due to bioleaching of arsenopyrite (Equation (1)) and decrease due to the formation of scorodite according to Equation (5) [39]:2H_3_AsO_4_ + Fe_2_(SO_4_)_3_ → 2FeAsO_4_↓ + 3H_2_SO_4_
(5)

In reactor 3, the concentration of arsenic in the medium increased slowly until the 25th day of the experiment, increasing from 1.4 to 2.3 g/L, after which it began to increase rapidly, reaching 10.3 g/L by the end of the experiment.

#### 3.1.2. Solid Residue Analysis

It was shown that the oxidation of sulfide sulfur was the highest in reactor 1; it was slightly lower in reactor 2, while in reactor 3 it was approximately two times lower (Table 3 and Table 4). The oxidation rate of arsenopyrite was almost equal in reactors 1 and 2, and insignificantly lower in reactor 3. At the same time, the rate of pyrite oxidation differed (Table 3 and Table 4).

This may be due to that arsenopyrite is more easily oxidized than pyrite, and when it is oxidized in a mixture of arsenopyrite and pyrite, electrochemical interactions occur between the minerals, which accelerate the oxidation of arsenopyrite [40,41]. In addition, it should be noted that the rapid oxidation of pyrite requires a higher Eh than arsenopyrite [41,42], while in reactor 3, Eh was lower throughout almost the entire experiment than in reactors 1 and 2 (Figure 1b). Thus, even less intense bio-oxidation of the concentrate in the control reactor made it possible to oxidize arsenopyrite at a rather high rate, while differences in the bio-oxidation activity influenced pyrite oxidation to a greater extent.

It should be noted that the mass yield of the solid residue of the concentrate bio-oxidation in reactors 1 and 2 was higher than in reactor 3, despite the more intense bio-oxidation. Obviously, this was due to the formation of calcium sulfate precipitate, which is formed when calcium carbonate is added into the medium containing sulfuric acid to maintain the pH in the reactor pulp:H_2_SO_4_ + CaCO_3_ → CaSO_4_↓ + CO_2_ + H_2_O(6)

Since the consumption of CaCO_3_ in reactors 1 and 2 was significantly higher, this could lead to an increase in the mass yield of the bio-oxidation residue. 

The phase analysis results were confirmed by XRD analysis (Table 5), which demonstrated the difference in the content of different minerals including pyrite and arsenopyrite and calcium sulfate-containing minerals in bio-oxidation residues obtained.

Thus, at 40 °C, both carbon dioxide and molasses significantly intensified bio-oxidation of the sulfide concentrate, which led to differences in the parameters of the pulp liquid phase as well as mineral oxidation rates (Table 2, Table 3, Table 4 and Table 5).

#### 3.1.3. Microbial Population Analysis

Analysis of the microbial population showed that changes in the total number of microorganisms corresponded to changes in the parameters of the liquid phase (Figure 2a).

On the 10th day, the number of microorganisms in the liquid phase of the pulp was lower than at the beginning of the experiment (1 × 10^8^ cells/mL and 0.5–0.6 × 10^7^ cells/mL, respectively). This may be explained by the fact that during the bio-oxidation of sulfide minerals, microorganisms at the beginning of the process are actively attached to mineral particles, which can lead to a decrease in the number of microorganisms in the liquid phase of the pulp at the initial stages of oxidation [43,44]. Moreover, a decrease in the microbial cell number might be caused by their lysis due to the inhibiting factors associated with a high pulp density: A high concentration of heavy metal ions, mechanical damage to cells with solid particles, saturation of the surface layers of bacteria with mineral particles that impede the entry of nutrients into the cell, and the release of exometabolites [45,46].

In reactors 1 and 2, microbial cell numbers in the liquid phase began to increase after 10 days of the bio-oxidation (Figure 2a, curves 1 and 2). In reactor 1, it reached a maximum on the 20th day (9.2 × 10^8^ cells/mL), after which it decreased (Figure 2a, curve 1). It should be noted that in reactor 1, the highest rate of iron and arsenic leaching, as well as a decrease in pH (Figure 1a,c,d, curve 1), was observed between 10 and 20 days, which corresponded to the interval in which the most rapid growth of the cell number in the liquid phase occurred. In reactor 2, the number of microorganisms began to increase after 10 days and grew until the end of the experiment. The growth rate of microbial cell number was maximum up to the 30th day, after which it decreased (Figure 2a, curve 2). In this case, the maximum rate of increase in cell number coincided with the maximum rate of iron leaching from the concentrate (Figure 1c,e curve 2), while the arsenic concentration rapidly increased in the range of 10–15 days, when no rapid increase in microbial cell number was observed in reactor 2 (Figure 1e, curve 2). 

In reactor 3, a relatively rapid increase in the number of microbial cells began after 20 days of bio-oxidation, but after 30 days, there was practically no increase in the number of cells (Figure 2a, curve 3). In reactor 3, the maximum leaching rate did not correspond to the maximum growth rate of microbial cell number but was observed after 30 days of bio-oxidation (Figure 1c,e, curve 2). This phenomenon may be due to the consumption of the intermediates of the concentrate oxidation (Fe^2+^, sulfur reduced compounds) accumulated over the first 20 days of the process by microorganisms in the period between 20 and 30 days of the experiment. It should be noted that the total number of microbial cells in control reactor 3 was 2–3 times lower than in reactors 1 and 2, where carbon dioxide and molasses were used as a carbon source, respectively (Figure 2a).

Thus, at 40 °C, the use of additional carbon sources led to an increase in the intensity of the concentrate bio-oxidation. Analysis of the effect of carbon sources on the number of microorganisms showed that the use of additional carbon sources led to an increase in the number of microorganisms in the reactor pulp. The introduction of additional carbon sources led to a reduction in the duration of the lag phase (in 1 and 2 reactors, active growth was observed after 10 days, while in control reactor 3 it began only after 20 days).

Molecular biological analysis of microbial population in the reactors revealed that the introduction of additional carbon sources also led to a change in the composition of the microbial communities of bioreactors. At 40 °C, representatives of the genera *Sulfobacillus*, *Acidithiobacillus*, *Acidiphilum*, *Acidiferrobacter*, *Leptospirillum*, *Ferroplasma*, *Cuniciliplasma*, and uncultivated archaea of the A-plasma group were identified in microbial communities. At the same time, the shares of representatives of genera *Sulfobacillus*, *Acidithiobacillus* (*Acidithiobacillus caldus*), *Leptospirillum*, and *Ferroplasma* were significant.

In reactor 1, on days 20, 30, and 40, the proportion of *Ferroplasma* archaea was significant (Figure 2b). Obviously, in this reactor, CO_2_ supply allowed mixotrophic archaea to adapt to growth in the pulp, despite the absence of organic matter, due to carbon dioxide fixation that promotes their growth.

On the 20th day, which corresponded to the maximum number of microorganisms, the share of *Acidithiobacillus caldus* bacteria was also very significant.

On days 30 and 40, when the total number of microorganisms decreased, the proportion of the bacteria of the genus *Sulfobacillus* and *Leptospirillum* increased, while the proportion of *Acidithiobacillus caldus* decreased. Probably, the period of a decrease in the total number of microbial cells could be accompanied by partial lysis of cells of the autotrophic sulfur oxidizer *Acidithiobacillus caldus*, which led both to a decrease in the proportion of sequences of the genus *Acidithiobacillus* and to an increase in the proportion of mixotrophic bacteria of the genus *Sulfobacillus* after 30 days, which can consume lysis products of autotrophic microorganisms.

The proportion of archaea was also significant in reactor 2 (Figure 2c). On the 20th day, the proportion of the sequences from the genus *Ferroplasma* accounted for over 90%.

On days 30 and 40, the proportion of bacteria *Acidithiobacillus caldus*, *Sulfobacillus*, and *Leptospirillum* increased. The predominance of the archea of the genus *Ferroplasma* in the microbial population of reactor 2 can be explained by the addition of molasses into the medium since representatives of the genus *Ferroplasma* is generally able to use organic matter as a source of carbon and energy [11].

Partial replacement of the archaea by the bacteria *A. caldus* and representatives of the genus *Leptospirillum* was observed on the 30th day of the experiment. By the end of the process, the share of *Ferroplasma* again increased due to the accumulation of exometabolites of autotrophic bacteria, which also may induce the growth of *Ferroplasmaceae* representatives [33]. It should be noted that up to 20 days the rate of iron leaching was relatively low, and after 20 days it increased. Possibly, this may indicate a potentially important role of the bacteria *Acidithiobacillus caldus*, *Sulfobacillus*, and *Leptospirillum* in pyrite leaching in comparison with archaea of the genus *Ferroplasma*.

It was not possible to obtain a sufficient amount of biomass from the pulp sample collected from reactor 3 on the 20th day to isolate DNA for analysis according to the methods used, which was obviously due to the low number of microorganisms (Figure 2a, curve 3). On the 30th day, the proportions of the sequences of bacteria *Sulfobacillus* and *Leptospirillum* and archaea of the genus *Ferroplasma* were significant. On the 40th day, the proportion of sequences of bacteria *Acidithiobacillus caldus* and *Leptospirillum*, as well as archaea of the genus *Ferroplasma* increased, while the proportion of the bacteria of the genus *Sulfobacillus* decreased (Figure 2d). 

### 3.2. Biooxidation under Thermophilic Conditons

The results of the experiment on bio-oxidation of the concentrate at 50 °C are shown in Figure 3 and Figure 4, as well as in Table 2, Table 3, Table 4 and Table 5.

#### 3.2.1. Liquid Phase Analysis

In contrast to the bio-oxidation at 40 °C, at 50 °C only the addition of carbon dioxide made it possible to intensify the bio-oxidation process, while the addition of molasses to the medium affected the bio-oxidation to a lesser extent, which was demonstrated both by the parameters of the liquid phase of the pulp and by the oxidation rate of sulfide minerals (Table 2, Table 3, Table 4 and Table 5). 

The pH of the pulp liquid phase in reactor 1 rapidly decreased from 1.40 to 0.70 on the 20th day. Then, the pH in the reactor gradually increased due to the addition of CaCO_3_ and by the end of the experiment it was 1.72 (Figure 3a, curve 1). In reactor 2, in which molasses was added to the medium, and in control reactor 3, the pH values decreased more slowly and it was possible to maintain it in the range 1.25–1.55 (Figure 3a, curves 2 and 3). In all reactors, CaCO_3_ was added to the slurry to maintain the pH. In reactor 1, the consumption of CaCO_3_ was 456 kg/t of concentrate, in reactor 2 it was 137 kg/t, while in control reactor 3, it comprised 102 kg/t. 

Changes in the Eh of the medium (Figure 3b), which reflect the activity of iron oxidation in the medium, corresponded to the changes in pH (Figure 3a). In reactor 1, Eh began to grow rapidly after 10 days and on the 12th day of bio-oxidation exceeded 700 mV, and then was in the range of 724–795 mV (Figure 3b, curve 1). In reactors 2 and 3, the Eh values increased slowly compared to reactor 1 and reached maximum values on days 35–37 (680–700 mV) (Figure 3b, curves 2 and 3), which was lower than the average Eh values in the reactor 1 (Figure 3b, curve 1).

The diagrams in Figure 3c,d show changes in the concentration of iron ions. Up to the 10th day of bio-oxidation, the total concentration of iron ions practically did not change in all reactors (Figure 3c, curves 1−3), while in all reactors the concentration of Fe^2+^ ions increased (Figure 3d, curves 1−3). After 9 days of bio-oxidation in reactor 1, the concentration of Fe^2+^ ions rapidly decreased and remained low until the end of the experiment (less than 1.00 g/L), exceeding 1.00 g/L only on days 20, 35, 37, and 40 (Figure 3d, curve 1). This indicates that during the first 10 days of the experiment, the bio-oxidation activity was low, and accumulation of ferrous iron ions, as in the experiment at 40 °C, occurred. After 10 days in reactor 1, the total concentration of iron ions Fe^3+^ and Fe^2+^ in the medium increased at a high rate, reaching a maximum (56.00 g/L) by the end of the experiment (Table 2, Figure 3c, curve 1). In reactors 2 and 3, ferrous iron in the medium was slowly oxidized up to 27–28 days, after which its concentration rapidly decreased, but still remained sufficiently high until the end of the experiment (Table 2, Figure 3d, curves 2 and 3). The total concentration of iron ions Fe^3+^ and Fe^2+^ in reactors 2 and 3 in the medium increased slowly and at the end of the experiment were 6–7 times lower than in reactor 1 (Table 2, Figure 3c, curves 2 and 3). 

The dynamics of changes in the concentration of arsenic in the medium were similar to the dynamics of changes in the concentration of iron ions (Figure 3d). In reactor 1, the arsenic concentration began to increase rapidly after 10 days of bio-oxidation, reaching a maximum (11.4 g/L) on the 15th day of bio-oxidation (Figure 1e, curve 1). Then the concentration of arsenic in the medium changed insignificantly (Figure 1e, curve 1). In reactors 2 and 3, the arsenic concentrations in the medium increased more slowly and by the end of the experiment were 4–6 times lower than in reactor 1. 

#### 3.2.2. Solid Residue Analysis

Table 3 and Table 4 summarize the results of the phase analysis of the solid phase. It was shown that the oxidation state of sulfide sulfur was the highest in reactor 1, as well as during bio-oxidation at 40 °C, while in reactors 2 and 3, the oxidation rate of sulfide sulfur was more than 3–5 times lower. 

The oxidation rate of pyrite in reactor 1 was the highest and higher than that in the experiment at 40 °C. Obviously, this may be due to that at a higher temperature the mineral was oxidized at a higher rate, and the use of carbon dioxide made it possible to create favorable conditions for bio-oxidation since microbial population promoting active oxidation of sulfide minerals formed under these conditions. 

At the same time, the oxidation rates of pyrite in reactors 2 and 3 were low (about 20% and 10%, respectively). Arsenopyrite oxidation rate was high in all reactors, but in reactor 1 it was higher. It should be noted that the mass yield of the solid residue of concentrate bio-oxidation in reactor 1 was lower, despite the greater amount of added calcium carbonate, while in reactors 2 and 3 it was close to 100%. The phase analysis results were confirmed by XRD analysis (Table 5). 

Thus, at 50 °C, the use of carbon dioxide significantly affected the rate of bio-oxidation, while the use of molasses did not significantly affect the bio-oxidation of the concentrate.

#### 3.2.3. Microbial Population Analysis

Analysis of the microbial population showed that changes in the total number of microorganisms generally corresponded to changes in the parameters of the liquid phase and the results of bio-oxidation of the concentrate. However, the number of microorganisms in reactors at 50 °C was several times lower than at 40 °C (Figure 4a). In reactor 1, the number of microbial cells in the liquid phase began to increase rapidly after 15 days and reached a maximum at 20–25 days (2.1–2.5 × 10^8^ cells/mL), after which it began to decrease (Figure 4a, curve 1). In reactors 2 and 3, the number of microorganisms began to increase after 20 days but was rather low (2.1 and 1.2 × 10^8^ cells/mL, respectively).

Molecular biological analysis revealed that the introduction of additional carbon sources into the bioreactor pulp also led to a change in the composition of the microbial communities of the bioreactors. At 50 °C, representatives of the genera *Sulfobacillus*, *Acidibacillus*, *Alicyclobacillus*, *Acidithiobacillus*, *Acidiphilum*, *Acidiferrobacter*, *Leptospirillum*, *Ferroplasma*, *Acidiplasma*, *Cuniciliplasma*, and uncultivated archaea A-plasma were revealed in microbial communities. The shares of representatives of bacteria *Sulfobacillus* and *Acidithiobacillus* (*Acidithiobacillus caldus*) and archaea of the genera *Ferroplasma* and *Acidiplasma*, as well as the uncultivated group A-plasma were significant, while proportions of the sequences of other groups were comparatively low. 

In reactor 1, on days 20, 30, and 40, there was a significant proportion of archaea from the genus *Acidiplasma* (95–25%) (Figure 4b). On the 20th day, which corresponded to the maximum number of microbial cells, the proportion of bacteria *Acidithiobacillus caldus* was also relatively high (5%). On days 30 and 40, during which the total number of microorganisms decreased, the proportion of the bacteria *Sulfobacillus* and *Acidithiobacillus caldus* increased, while share of the archaea *Ferroplasma* and A-plasma decreased. It should be noted that up to 20 days the rate of iron and arsenic leaching was maximum, and after 20 days the concentration of arsenic in the medium stabilized, while the rate of iron leaching decreased, but the iron concentration continued to increase. It is possible that in the period from 10 to 20 days arsenopyrite and the relatively easily oxidized fraction of pyrite were intensively oxidized, while after 20 days further oxidation of the concentrate minerals occurred, which may indicate a potentially large role of the bacteria *Acidithiobacillus caldus* and *Sulfobacillus*, which share has increased significantly, in the oxidation of the most resistant minerals. 

In reactor 2, the proportion of archaea of the genus *Acidiplasma* was significant (Figure 4c). On days 20, 30, and 40, the proportion of the sequences of the archaea of the genus *Acidiplasma* accounted for 90, 77, and 82%, respectively. The predominance in the microbial community of reactor 2 archaea of the genus *Acidiplasma* can be explained by the introduction of molasses into the medium since representatives of this genus are obviously iron-oxidizing mixotroph and their growth depends on the presence of an organic carbon source in the medium. Previously, it was shown that the number of archaea of the genus *Acidiplasma* in bio-oxidation reactor communities increased significantly when an organic carbon source was introduced into the environment [20]. 

It was not possible to obtain sufficient biomass from the pulp sample collected from reactor 3 on the 20th day to isolate DNA for analysis, which was obviously due to the low number of microorganisms, as well as in the experiment at 40 °C (Figure 4a, curve 3). On days 30 and 40, the proportions of the sequences of bacteria *Sulfobacillus* and bacteria *Acidithiobacillus caldus* and archaea of the genus *Acidiplasma* were significant. 

Analysis of the results of the analysis of microbial communities of bio-oxidation reactors at 50 °C and their comparison with the results of bio-oxidation showed that the addition of molasses led to a relatively insignificant increase in the total number of microorganisms and to an increase in the proportion of archaea of the genus *Acidiplasma*, which did not lead to an intensification of the bio-oxidation process.

## 4. Discussion

The results obtained in this work showed that the use of both inorganic (carbon dioxide) and organic (molasses) additional carbon sources made it possible to intensify the growth of microorganisms and the leaching process. It was demonstrated that the use of both investigated carbon sources led to an increase in the number of microorganisms in the liquid phase of the bioreactor pulp. At the same time, the use of carbon dioxide led to a more rapid increase in the number of microorganisms and to a more significant reduction in the lag phase, which was accompanied by a more rapid leaching of iron from the concentrate.

At 40 °C, the use of molasses also made it possible to increase the rate of bio-oxidation of the concentrate; however, it should be noted that the rate of iron leaching in this case increased more slowly than when using carbon dioxide. The leaching rate of arsenic was high using both carbon sources. At 50 °C, only carbon dioxide addition made it possible to significantly increase the rate of sulfide concentrate bio-oxidation.

Obviously, the effect of carbon sources on the process of bio-oxidation of sulfide concentrate was due to their effect on microbial communities that developed in bio-oxidation reactors. At 40 °C, the total number of microorganisms when using additional carbon sources was several times higher than in the control experiment.

At the same time, differences were observed not only in the number of microorganisms, but also in the composition of microbial populations. In reactors 1 and 2, the proportion of archaea of the genus *Ferroplasma* was higher than in the control reactor. In reactor 1, where carbon dioxide was used, there was a relatively high proportion of bacteria *A. caldus* (at the beginning of the process) and *Sulfobacillus* (at the end of the process). It is likely that the supply of carbon dioxide to the pulp allowed autotrophic bacteria *A. caldus* to develop at the initial stage of the process, and the accumulation of its metabolites and, possibly, cell lysis products in the medium led to an increase in the proportion of mixo- and heterotrophic iron-oxidizing microorganisms in the population at subsequent stages.

When adding molasses to the medium, the proportion of archaea sequences of the genus *Ferroplasma* was the highest, but then the proportion of acidophilic bacteria *Leptospirillum*, *A. caldus*, and *Sulfobacillus* also increased. Thus, despite the differences in the composition of microbial communities revealed under different conditions, changes in the composition of microbial populations were observed during bio-oxidation, and it is obvious that different microorganisms could predominate at different stages of bio-oxidation. Interestingly, when an organic carbon source was added to the medium at the end of the process, autotrophic microorganisms developed. It is likely that at the beginning of the process, there was an accumulation of components in the oxidation products that contributed to their development. For example, the predominance of iron-oxidizing archaea of the genus *Ferroplasma*, which require an organic carbon source, in reactor 2 could lead to the accumulation of elemental sulfur in the solid phase during the bio-oxidation of arsenopyrite [12,47], that resulted in an increase in the proportion of autotrophic sulfur-oxidizing bacteria *A. caldus*. In addition, in reactors 1 and 2, at the end of the experiment, there was an increase in the proportion of bacteria *Leptospirillum* and *Sulfobacillus*, which can play an important role in the oxidation of pyrite [42,48].

It should be noted that in reactor 3, where the source of carbon for the microbial community could only be carbon dioxide, which is introduced with the aeration, there was a significant proportion of acidophilic iron and sulfur-oxidizing bacteria. Despite this, the intensity of bio-oxidation in the control reactor 3 was the lowest, which is obviously due to the low total number of microorganisms.

At 50 °C, the use of carbon dioxide significantly affected the rate of bio-oxidation, while the effect of molasses on the bio-oxidation of the concentrate was comparatively low. In contrast to bio-oxidation at 40 °C, at 50 °C the proportions of the genera *Leptospirillum* and *Ferroplasma* were low, while archaea of the genus *Acidiplasma* were one of the predominant groups in all reactors that may be explained by higher temperature optimum of the representatives of genus *Acidiplasma* in comparison to *Leptospirillum* and *Ferroplasma* representatives [1,49]. 

In reactor 1, where carbon dioxide was used, the proportion of the bacteria *Sulfobacillus* and *Acidithiobacillus caldus* was high on days 30 and 40, despite *Acidiplasma* sequences being predominant at the beginning of the process. 

In reactor 2, where organic carbon source was used, the proportion of archaea of the genus *Acidiplasma* was the highest that may be explained by its dependence on the organic carbon source. It is corresponded to the data obtained in our previous work, as it was shown that the number of archaea of the genus *Acidiplasma* in bio-oxidation reactor increased significantly when an organic carbon source (yeast extract) was used to increase the bio-oxidation rate [20]. 

In reactor 3, where no additional source of carbon for the microbial population was used bio-oxidation rate was the lowest, which is obviously due to the lowest total number of microorganisms. It may suggest that at higher temperatures, the activity of the microbial population depended on the presence of additional carbon sources to a greater extent. This fact may be explained by the decrease in CO_2_ solubility with an increasing temperature that led to the deficiency in dissolved CO_2_ at 50 °C in the absence of additional carbon sources.

It should be noted that a long lag phase was observed in all experiments (10–25 days). This may be due to the inhibitory effect of high pulp density (20% solid) on the cells of the main groups of bacteria that oxidize sulfide minerals in this work [45]. The use of additional carbon sources made it possible to reduce the duration of the lag phase.

Thus, it was shown that the use of various carbon sources made it possible to significantly increase the intensity of bio-oxidation of the sulfide concentrate since it influenced the number of microorganisms and the composition of microbial populations. Obviously, in the processes of bio-oxidation of sulfide concentrates, the availability of carbon for auto-, mixo-, and heterotrophic microorganisms can be a limiting factor, and the study of the effect of various carbon sources on the intensity of the bio-oxidation process is a promising direction for the development of biohydrometallurgical technologies. 

Indeed, the use of additional carbon sources may lead to additional costs to equip the reactors with additional nutrients (both carbon dioxide and organic nutrients). At the same time, the application of additional carbon nutrition may allow resolving a key issue in industrial-scale bio-oxidation processes—inhibition of bio-oxidation activity due to the overheating of bioleach reactor pulp. In our work, it was shown that bio-oxidation may be performed efficiently both at 40 and 50 °C when using carbon dioxide supply. As carbon dioxide was shown to be profitably used in industrial-scale processes [6], it may be also successfully used to prevent the negative effects of overheating as well as to decrease costs when applying cooling systems since CO_2_ supply provided effective bio-oxidation under both mesophilic and thermophilic conditions.

In the present work, the experiments were performed in a batch mode, while more reliable results—from the perspective of modeling industrial-scale processes—may be obtained in continuous laboratory trials. Nevertheless, the results obtained provide strong evidence of the hypothesis that additional carbon nutrition can significantly affect bio-oxidation efficiency and decrease the negative effect of temperature rise. Thus, based on the results of the present work, further continuous laboratory and pilot tests may be designed. 

## 5. Conclusions

Based on the findings through the bath laboratory tests, it was shown that the use of various carbon sources made it possible to increase the intensity of pyrite-arsenopyrite concentrate bio-oxidation since it affected microbial populations performing the process. The application of carbon dioxide provided more effective bio-oxidation at different temperatures (40 and 50 °C) in comparison to the control tests. Thus, carbon dioxide supply may be proposed as an effective approach to minimize the negative effects of temperature rise during bio-oxidation processes as well as to decrease costs for cooling of industrial-scale reactors.

## Figures and Tables

**Figure 1 microorganisms-09-02350-f001:**
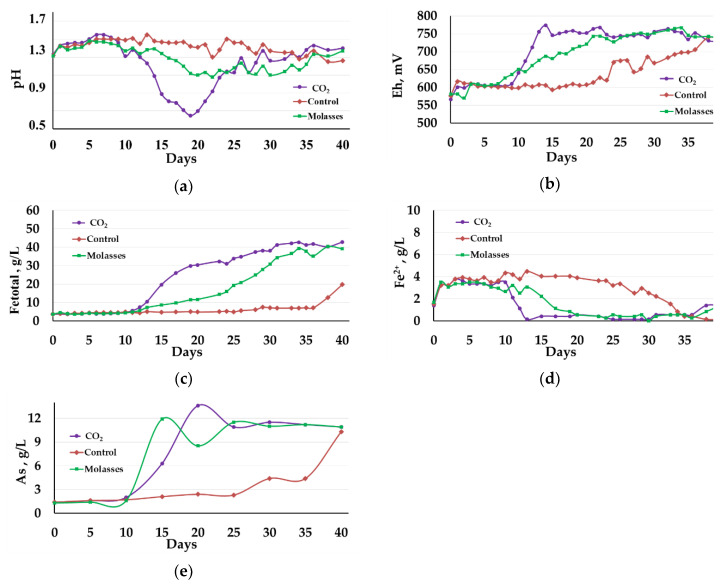
Changes in the liquid phase parameters during the bio-oxidation of the concentrate at 40 °C: (**a**)—pH; (**b**)—Eh; (**c**)—total concentration of Fe^3+^ and Fe^2+^ ions (g/L); (**d**)—concentration of Fe^2+^ ions (g/L); (**e**)—concentration of arsenic (g/L).

**Figure 2 microorganisms-09-02350-f002:**
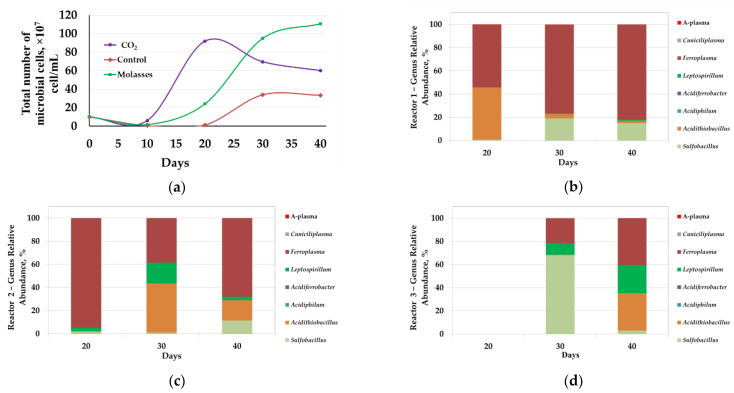
Analysis of microbial communities of bio-oxidation reactors (40 °C): (**a**)—total number of microbial cells (10^7^ cells/mL); results of molecular biological analysis of microbial populations: (**b**)—reactor 1 (CO_2_); (**c**)—reactor 2 (molasses); (**d**)—reactor 3 (control).

**Figure 3 microorganisms-09-02350-f003:**
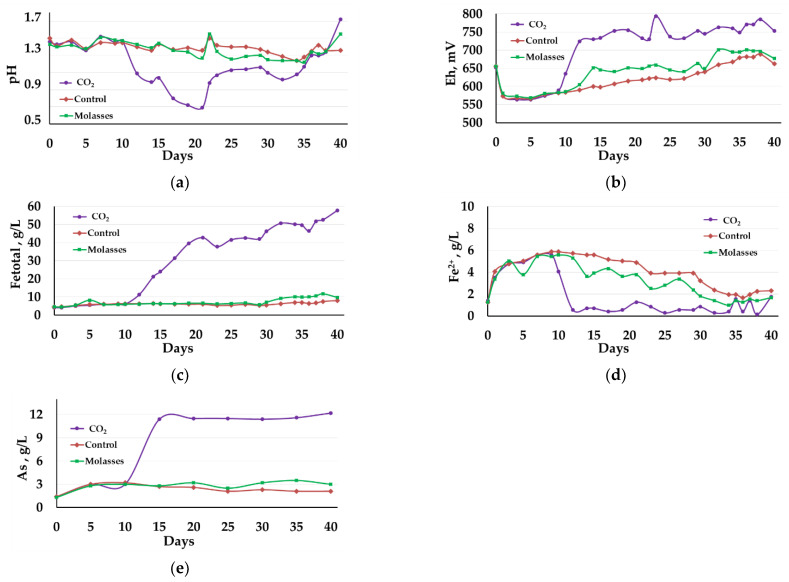
Changes in the liquid phase parameters during the bio-oxidation of the concentrate at 50 °C: (**a**)—pH; (**b**)—Eh; (**c**)—total concentration of Fe^3+^ and Fe^2+^ ions (g/L); (**d**)—concentration of Fe^2+^ ions (g/L); (**e**)—concentration of arsenic (g/L).

**Figure 4 microorganisms-09-02350-f004:**
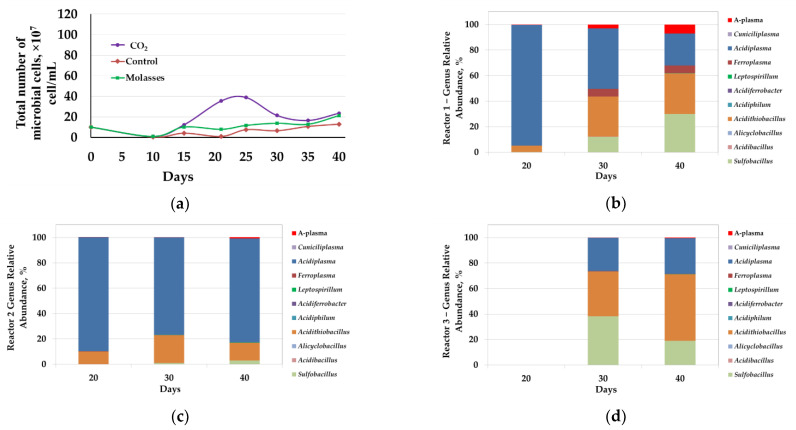
Analysis of microbial communities of bio-oxidation reactors (50 °C): (**a**)—total number of microbial cells (10^7^ cells/mL); results of molecular biological analysis of microbial populations: (**b**)—reactor 1 (CO_2_); (**c**)—reactor 2 (molasses); (**d**)—reactor 3 (control).

**Table 1 microorganisms-09-02350-t001:** Chemical composition of the concentrate.

Component	Content, %
Fe_tot_	31.8
Fe_s_	29.1
S_tot_	34.7
S_s_	34.4
S_sulfate_	0.2
S^0^	0.1
As_tot_	6.9
As_s_	6.5
Au, g/t	45.5

**Table 2 microorganisms-09-02350-t002:** Parameters of the liquid phase of the pulp at the end of the experiment (on the 40th day).

Temperature	Carbon Source	pH	Eh	Fe^3+^, g/L	Fe^2+^, g/L	As, g/L	Microbial Cell Number, × 10^7^ Cell/mL
40 °C	CO_2_	1.40	730	41.4	1.4	10.9	60
	Molasses	1.37	736	37.8	1.4	10.9	110
	Control	1.26	730	19.6	0.1	10.3	33
50 °C	CO_2_	1.72	753	56.0	1.8	12.0	24
	Molasses	1.55	667	8.1	1.7	3.0	22
	Control	1.36	662	5.7	2.3	2.1	12

**Table 3 microorganisms-09-02350-t003:** Chemical composition of bio-oxidation residues.

Mineral	Content, %					
	40 °C			50 °C		
	Reactor 1	Reactor 2	Reactor 3	Reactor 1	Reactor 2	Reactor 3
Fe_tot_	7.20	8.50	24.50	3.90	24.50	26.10
Fe_s_	8.62	16.28	27.00	1.60	20.37	22.57
S_tot_	20.10	19.90	31.40	16.20	28.50	29.90
S_s_	7.60	8.80	27.40	2.30	24.50	26.50
S_sulfate_	12.10	10.70	3.60	13.70	3.50	2.90
S^0^	0.40	0.40	0.40	0.10	0.50	0.50
As_tot_	0.21	0.26	1.20	0.85	5.30	5.85
As_s_	0.14	0.16	0.57	0.03	0.41	0.47

**Table 4 microorganisms-09-02350-t004:** Oxidation of sulfide minerals and sulfide sulfur after 40 days of bio-oxidation.

Temperature	Carbon Source	Yield, %	Oxidation Rate, %		
			S_S_	Pyrite	Arsenopyrite
40 °C	CO_2_	85.3	81	77	98
	Molasses	85.9	78	73	98
	Control	78.7	37	27	93
50 °C	CO_2_	73.7	95	94	99
	Molasses	98.7	29	21	94
	Control	103.9	19	10	92

**Table 5 microorganisms-09-02350-t005:** Mineral composition of the concentrate and bio-oxidation residues (XRD).

Mineral	Content, %						
	Concentrate	40 °C			50 °C		
		Reactor 1	Reactor 2	Reactor 3	Reactor 1	Reactor 2	Reactor 3
Pyrite (FeS_2_)	48	10	10	30	<1	30	35
Arsenopyrite (FeAsS)	9	<1	<1	<1	<1	<1	<1
Quartz (SiO_2_)	25	20	15	20	30	35	35
Mica	18	10	10	5	15	25	20
Gypsum (CaSO_4_·2H_2_O)	0	55	65	45	15	0	0
Bassanite (CaSO_4_·0.5H_2_O)	0	0	0	0	40	10	10
Anhydrite (CaSO_4_)	0	5	0	0	0	0	0

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
