# Peer review of "Effect of Carbon Sources on Pyrite-Arsenopyrite Concentrate Bio-oxidation and Growth of Microbial Population in Stirred Tank Reactors"

_microorganisms, 2021, doi:10.3390/microorganisms9112350_

Round 1
Reviewer 1 Report
General comment for the introduction. In this section, the extensive discussion of previous research should be avoided. Therefore, the introduction could be shortened resulting in smoother for readers.
Lines 32-35, 41-42, and 62-66. You wrote that this process has an economic advantage. But, in my experience, aeration is often associated with high costs. Also, the fact the reaction requires a high temperature, a method for keeping the temperature constant should be costly too. Finally, providing carbon dioxide is not economic. Please, clarify.
Line 156. So, did you use an artificially contaminated solution?
Line 163. Why not in a continuous mode?
Lines 164-165. How did you ensure the mesophilic and thermophilic conditions? Did you wrap the bioreactor with an aluminum folio to avoiding photooxidation?
Lines 167-170. Why did you choose to use such nutrients? Please, add a reference.
Line 170. Pure sulfuric acid?
Lines 183-203. This part should be included in a subsection called analysis and sampling. The previous could be called experimental setup.
General comment. The statistical analysis should be provided in the materials and methods section, and all the results should be statistically compared.
Results. Also here, the various aspect analyzed could be divided into different sub-section.
Figure 1. I don't see error bars. Did you performer the analysis in single or duplicate?
Figure 2d. I don't see the column relative to 20 °C. Also, the phylogenetic tree could be included in the revised version.
Discussion. It would be better to clarify the differences in terms of efficiency between the conditions.
The conclusion section should be included separate from the discussion in a short section of 150 words at maximum, by reporting main results and future perspectives.
Author Response
Point 1. General comment for the introduction. In this section, the extensive discussion of previous research should be avoided. Therefore, the introduction could be shortened resulting in smoother for readers.
Response 1:
The introduction has been shortened.
Point 2. Lines 32-35, 41-42, and 62-66. You wrote that this process has an economic advantage. But, in my experience, aeration is often associated with high costs. Also, the fact the reaction requires a high temperature, a method for keeping the temperature constant should be costly too. Finally, providing carbon dioxide is not economic. Please, clarify.
Response 2:
2.1. Indeed, both aeration and carbon dioxide supply constitutes sufficient part of operational cost of gold producing plants including those applying biooxidation technologies. It should be noted that all technologies used to process refractory sulfide concentrates (biooxidation, pressure oxidation, atmospheric oxidation, roasting) require intensive aeration with atmosphere air and, in some cases, with oxygen. Despite this, all these technologies are successfully used in industrial practice. Thus, aeration does not impede application of these technologies in industrial scale.
2.2. In the case of industrial scale stirred tank biooxidation reactors, keeping the high temperature is not an issue. In contrast, biooxidation of sulfide minerals in industrial scale reactors cause the release of heat. Therefore, industrial scale biooxidation reactors should be equipped with cooling systems to maintain required temperature (see references on Lines 49-59). The exploitation of cooling system is associated with high costs both due to operational costs and needed cooling system reparation. These costs are usually covered by the profit that has been proved by application of biooxidation technologies for treatment of gold-bearing concetrates worldwide. In the same, the reduction of the costs for cooling systems is relevant issue for biooxidation plants especially in terms of reducing energy consumption.
2.3. In our article, we considered application of additional carbon source as possible approach to avoid inhibition of biooxidation activity due to temperature rise. For this purpose, we proposed use two different carbon sources. Indeed, the use of additional carbon sources requires additional costs. Based on the results obtained, we cannot directly evaluate economic effect of the approach proposed as this require further studies (including long-term trials and engineering calculations). In the same, the approach proposed seems to be promising as in some cases carbon dioxide may be used in biooxidation practice as was shown by the developers of BIOX technology (lines 84–89, reference 6).
Additional sentences with explanations regarding point 2 were included in the discussion to clarify possible economic valuation of the results (lines 567-583).
Point 3. Line 156. So, did you use an artificially contaminated solution?
Response 3:
We did not use any contaminated solutions in our experiments. We have used mineral nutrient medium and mineral concentrate obtained from natural sulfide gold-bearing ore.
Point 4. Line 163. Why not in a continuous mode?
Response 4:
Indeed, trials in a continuous mode are best approach to evaluate possible effects of different factors on the processes used in industrial scale. In the time, experiments in a continuous mode are labor and time-consuming. Therefore, experiments in batch mode are always required to design and perform continuous experiments. In the present work, we performed batch experiments and obtained the results, which have been used to plan further experiments.
Additional sentence with explanation corresponding to the point 4 was included in the discussion (lines 567-583).
Point 5. Lines 164-165. How did you ensure the mesophilic and thermophilic conditions? Did you wrap the bioreactor with an aluminum folio to avoiding photooxidation?
Response 5:
5.1. Temperature in the reactors was maintained using ELMI TW-2.03 circulating water baths and U- U-shaped titanium heat exchangers, which made it possible to maintain pulp temperature. As temperature in the bath can differ from that in the reactor, temperature in the reactors was also controlled manually two times daily using thermometers. Temperature fluctuations did not exceeded ±1°Ð¡ during the experiments.
Additional sentence with explanation corresponding to the point 5 was included in the methods description (line 124).
5.2. The reactors were not wrapped with an aluminum folio as we did not perform special measures against photooxidation. In first, photooxidation cannot have significant effect in comparison to pyrite and arsenopyrite biooxidation by aerobic acidophilic microorganisms. In second, the reactors were operated in dark room without constant lighting and subjected to the bright light few hours daily during the sampling and sample analysis. In third, biooxidation pulp is opaque suspension, which cannot shine through, while photoxidation may play role, for example, in shallow light permeable waters. The reactors are made of thick glass that does not transmit UV.
Point 6. Lines 167-170. Why did you choose to use such nutrients? Please, add a reference.
Response 6:
We used this nutrient media in our previous works on biooxidation and it was shown that it provide growth of microorganisms and biooxidation.
Corresponding references [26] and [34] were included (lines 129-131).
Point 7. Line 170. Pure sulfuric acid?
Response 7:
Yes, chemically pure 98% sulphuric acid was used (line 130).
Point 8. Lines 183-203. This part should be included in a subsection called analysis and sampling. The previous could be called experimental setup.
Response 8:
Additional sub-section was included.
Point 9. General comment. The statistical analysis should be provided in the materials and methods section, and all the results should be statistically compared.
Response 9:
Additional sub-section was included.
Point 10. Results. Also here, the various aspect analyzed could be divided into different sub-section.
Response 10:
Additional sub-section was included.
Point 11. Figure 1. I don't see error bars. Did you performer the analysis in single or duplicate?
Response 11:
Experiments under all conditions studied were performed in single as long term labor-consuming experiments were performed using 6 large-volume laboratory scale reactors. In the same time, chemical analyses were performed in duplicate. Deviations in most analyses did not exceed 0.1 g/L for both iron and arsenic, while in some points both measurements were equal. Therefore, error bars were not places in Figure 1.
Point 12. Figure 2d. I don't see the column relative to 20 °C. Also, the phylogenetic tree could be included in the revised version.
Response 12:
Figures 2d and 4d do not contain the columns on the 20th day. Corresponding explanations are in the text:
Line 353: It was not possible to obtain a sufficient amount of biomass from the pulp sample collected from reactor 3 on the 20th day to isolate DNA for analysis according to the methods used, which was obviously due to the low number of microorganisms (Fig. 2a, curve 3).
Line 474: It was not possible to obtain sufficient biomass from the pulp sample collected from reactor 3 on the 20th day to isolate DNA for analysis, which was obviously due to the low number of microorganisms, as well as in the experiment at 40°C (Fig. 4a, curve 3).
The results of NGS sequencing are often not shown as phylogenetic trees as NGS provides a high number of short sequences, which always have some misleading single nucleotide substitutions due to Illumina sequencing features. Therefore, these sequences are usually included in OTUs, which are identified at genus and sometimes at species levels, but usually do not used to build phylogenetic trees.
Point 13. Discussion. It would be better to clarify the differences in terms of efficiency between the conditions.
Response 13:
The additional sentences were included (lines 567-583)..
Point 14. The conclusion section should be included separate from the discussion in a short section of 150 words at maximum, by reporting main results and future perspectives.
Response 14:
Additional section was included.

Reviewer 2 Report
General comments
The manuscript presented sufficient data to prove the effect of carbon sources on bioleaching of pyrite-arsenopyrite concentrate. High-throughput sequencing was also applied to observe the change of microbial population during the leaching process. The obtained data deserve to be published. However, the current manuscript was limited by the awkward presentation and organization. It requires a thorough improvement before publication.
Specific comments:
- The introduction is too long, it should be concise.
- Materials and Methods should be separated into subsection for clearer presentation.
- Results should be separated into subsection for clearer presentation.
- Awkward paragraph separation was found throughout the manuscript. It should be improved by a native English speaker.
- Fig. 1, 2, 3, and 4: Please add the legend for each paragraph, do not mark 1, 2, and 3. It is ambiguous.
- Fig. 2 and 4: "Proportion of the 16S rRNA sequence in the population" should be replaced by Genus relative abundance (%).
- Fig 2 and 4: A-plasma should not be italicized.
- Even though a separate conclusion section is not required but a paragraph representing a conclusion at the end of discussion section is required.
Author Response
REVIEWER 2
Point 1. The manuscript presented sufficient data to prove the effect of carbon sources on bioleaching of pyrite-arsenopyrite concentrate. High-throughput sequencing was also applied to observe the change of microbial population during the leaching process. The obtained data deserve to be published. However, the current manuscript was limited by the awkward presentation and organization. It requires a thorough improvement before publication.
Response 1:
The manuscript was modified according to the reviewer commentaries.
Point 2. The introduction is too long, it should be concise.
Response 2:
The introduction has been shortened.
Point 3. Materials and Methods should be separated into subsection for clearer presentation.
Response 3:
Additional sub-sections were included.
Point 4. Results should be separated into subsection for clearer presentation.
Response 4:
Additional sub-sections were included.
Point 5. Awkward paragraph separation was found throughout the manuscript. It should be improved by a native English speaker.
Response 5:
The paragraphs were re-arranged.
Point 6. Fig. 1, 2, 3, and 4: Please add the legend for each paragraph, do not mark 1, 2, and 3. It is ambiguous.
Response 6:
The legends were modified.
Point 7. Fig. 2 and 4: "Proportion of the 16S rRNA sequence in the population" should be replaced by Genus relative abundance (%).
Response 7:
Figure captions were modified.
Point 8. Fig 2 and 4: A-plasma should not be italicized.
Response 8:
Italics was removed.
Point 9. Even though a separate conclusion section is not required but a paragraph representing a conclusion at the end of discussion section is required.
Response 9:
Additional section was included.

Round 2
Reviewer 1 Report
The manuscript achieved a sufficient quality for publication.
Reviewer 2 Report
The manuscript was sufficiently improved and can be published in the present form.